

# The relationship between vigilance capacity and physical exercise: a mixed-effects multistudy analysis

Daniel Sanabria[1,2], Antonio Luque-Casado[3], José C. Perales[1,2], Rafael Ballester[4], Luis F. Ciria[1,2], Florentino Huertas[4] and Pandelis Perakakis[5]

[1] Department of Experimental Psychology, University of Granada, Granada, Spain
[2] Mind, Brain and Behavior Research Center (CIMCYC), University of Granada, Granada, Spain
[3] Center for Sport Studies, Rey Juan Carlos University, Madrid, Spain
[4] Faculty of Physical Education & Sport Sciences, Catholic University of Valencia "San Vicente Martir", Valencia, Spain
[5] Department of Psychology, Universidad Loyola Andalucía, Seville, Spain

Corresponding author
Antonio Luque-Casado,
antonio.luque@urjc.es

## ABSTRACT

A substantial body of work has depicted a positive association between physical exercise and cognition, although the key factors driving that link are still a matter of scientific debate. Here, we aimed to contribute further to that topic by pooling the data from seven studies ($N = 361$) conducted by our research group to examine whether cardiovascular fitness ($VO_2$), sport type participation (externally-paced (e.g., football or basketball) and self-paced (e.g., triathlon or track and field athletes) vs. sedentary), or both, are crucial factors to explain the association between the regular practice of exercise and vigilance capacity. We controlled for relevant variables such as age and the method of $VO_2$ estimation. The Psychomotor Vigilance Task was used to measure vigilance performance by means of reaction time (RT). The results showed that externally-paced sport practice (e.g., football) resulted in significantly shorter RT compared to self-paced sport (e.g., triathlon) and sedentary condition, depicting larger effects in children and adolescents than in adults. Further analyses revealed no significant effect of cardiovascular fitness and self-paced sport practice, in comparison to the sedentary condition, on RT. Our data point to the relevance of considering the type of sport practice over and above the level of cardiovascular fitness as crucial factor to explain the positive association between the regular practice of exercise and vigilance capacity.

## INTRODUCTION

The current trend toward a sedentary lifestyle in modern societies clashes with the human natural tendency to be physically active (*Andersen, Mota & Pietro, 2016*; *Blair, 2009*). This pervasive lack of regular physical activity has been related to numerous chronic physical and mental diseases and, relevant to this article, to suboptimal cognitive functioning (*Booth, Roberts & Laye, 2012*). Indeed, there is a substantial body of work

depicting a positive association between regular practice of physical activity and cognition (*Gomez-Pinilla & Hillman, 2013*; *McMorris, 2016*). The key factors driving that link, however, are still a matter of scientific debate (*Stillman et al., 2016*). The aim of this brief report was to further contribute to that topic by testing the role of two critical variables, cardiovascular fitness and sport type participation, on the association between the regular practice of exercise and the level of vigilance (i.e., the ability to stay focused, and to detect and respond efficiently to target stimuli in order to attain the goals of the task). The interest in the study of vigilance (or sustained attention) was motivated by its crucial role in general cognitive capacities and to achieve optimal performance in many daylife activities (e.g., driving, attending to an academic lesson, etc.) (*Larue, Rakotonirainy & Pettitt, 2011*). To accomplish the objective of the present study, we performed a mixed-effects analysis including data from seven studies conducted by our research group (total $N = 361$) that used the same reaction time (RT) measure from the Psychomotor Vigilance Task (PVT) (*Luque-Casado et al., 2013*).

Exercise is typically defined as physical activity performed in a structured, planned and repetitive manner (*Caspersen, Powell & Christenson, 1985*). Enhanced physical or cardiovascular fitness is one of the main consequences of the regular practice of physical exercise that has been related to cognitive performance (*Gomez-Pinilla & Hillman, 2013*; *Åberg et al., 2009*; *Tarumi & Zhang, 2015*; *Voss et al., 2016*). In the particular case of vigilance, our studies showed that higher-fit individuals outperformed lower-fit individuals in the PVT (*Luque-Casado et al., 2013*; *Ciria et al., 2017*; *Luque-Casado et al., 2016b*). The evidence comes from both behavioral (RT) and electrophysiological measures (event related brain and cardiac potentials). The outcome of this research appears to support the cardiovascular fitness hypothesis, by virtue of which, physiological adaptations (e.g., increased $VO_2$, increased brain-derived neurotrophic factor (BDNF), etc.) induced by regular exercise are assumed to be responsible for observed cognitive improvements (*North, McCullagh & Tran, 1990*; *Voss, 2016*).

Given the variety of exercise-related contexts available, however, exercise practice is far more than just a way for enhancing cardiovascular fitness (*Hartman et al., 2017*). In fact, given its inherent and varying perceptual and cognitive demands, it is expected that cognitive enhancement would follow sustained practice (*Alesi et al., 2015*). For instance, optimal performance in football or basketball requires, together with a sufficiently good level of fitness, rapid adaptation and response to the constantly varying exercise environment (i.e., they are instances of externally-paced activities). In contrast, endurance cycling or long distance running involve self-regulation of the effort in a relatively consistent and predictable environment (i.e., they are instances of self-paced exercise). It is therefore reasonable to expect that any cognitive improvement related to the regular practice of exercise would depend on the particular activity and the associated cognitive demands. Our research on vigilance (*Ballester et al., 2017*) has taken that possibility into consideration, reporting that individuals (children) who practice externally-paced exercise regularly outperform those who practice self-paced exercise (i.e., they show shorter RTs in absence of differences in cardiovascular fitness between the two groups of athletes). These findings are consistent with the "cognitive skill" hypothesis

**Table 1 Summary of sample and task characteristics, VO₂ estimation methods and results of the studies included in the analysis.**

| Study | Sample size | Age range | Sex | Groups and sport type | PVT paradigm peculiarities | VO₂ estimation methods | Results (RT) |
|---|---|---|---|---|---|---|---|
| Study 1 *Luque-Casado et al. (2013)* | N = 26 | 17–29 | M | SP = 13 S = 13 | Duration: 10 min Stimuli: RC | A | SP < S |
| Study 2 *Ballester et al. (2015)* | N = 75 | 13–14 | M & F | EP = 39 (15 females) S = 36 (18 females) | Duration: 9 min Stimuli: GP | B | EP < S |
| Study 3 Luque-Casado & Sanabria (unpublished data, 2014) | N = 41 | 40–50 | M | SP = 22 S = 19 | Duration: 12 min Stimuli: GP | C | SP = S |
| Study 4 *Luque-Casado et al. (2016b)* | N = 50 | 18–32 | M | SP = 25 S = 25 | Duration: 60 min Stimuli: RC | C | SP < S (*) |
| Study 5 *Ciria et al. (2017)* | N = 43 | 18–23 | M & F | SP = 21 (10 females) S = 22 (11 females) | Duration: 5 min Stimuli: RC | C | SP < S |
| Study 6 *Ballester et al. (2017)* | N = 60 | 10–11 | M & F | EP = 20 (eight females) SP = 20 (12 females) S = 20 (10 females) | Duration: 9 min Stimuli: GP | B | EP < S EP < SP SP = S |
| Study 7 *Ballester et al. (2019)* | N = 66 | 18–37 | M & F | EP = 22 (10 females) SP = 22 (eight females) S = 22 (12 Females) | Duration: 9 min Stimuli: GP | A | EP < S EP = SP SP = S |

Notes:

Sex: M = Male, F = Female; Sport type: SP = Self-paced, EP =Externally-paced, S = Sedentary; Stimuli: RC = red circumference, GP = gabor patch; VO₂ (ml · kg⁻¹ · min⁻¹) estimation methods: A = estimation of the $VO_{2max}$ from the maximum power output in a maximal incremental cycle-ergometer test, B = estimation of the $VO_{2max}$ from the Léger Multi-stage fitness test and C = direct measure of oxygen uptake at the ventilatory anaerobic threshold (VAT; $VO_{2\ at\ VAT}$) in a submaximal incremental cycle-ergometer test.

* The shorter RT showed by SP group was limited to the first 36′ of the task.

(*Mann et al., 2007*; *Voss et al., 2010*) whereby the learning of basic cognitive abilities through practice of one particular activity can be transferred to other domains. This result may in turn jeopardize the notion that cardiovascular fitness is critical for differences in vigilance performance to occur as a function of exercise practice.

At this point, our research has not hitherto provided a clear answer to the issue of whether cardiovascular fitness, sport type participation, or both, are among the key factors determining the relationship between regular practice of exercise and vigilance performance. For this reason, we have decided to perform a mixed-effects multistudy analysis on data from seven studies (six published and one non-published) that our research group has conducted so far on this topic (Table 1) involving a total of 361 participants.

The use of the same task (the PVT) in all studies enabled us to incorporate the raw RT data, which we believe represents an advantage over the use of standardized effect sizes that are included in typical meta-analytical reports. To maintain homogeneity between studies, we only analyse RT from the first 5 min of the task, that corresponds to the shortest version of the PVT we have used so far. Note that the 5 min version of the task has been reported to be a reliable tool to assess vigilance performance (*Loh et al., 2004*). Importantly, apart from including cardiovascular fitness and sport type participation as main variables of interest in the analysis, we controlled for age, sex and the method of cardiovascular fitness estimation (that differed between studies).

In brief, this report aims to advance our current understanding of the research linking physical exercise and cognition, with a focus on vigilance. The details of the analysis

are reported below and the raw data and the R scripts can be downloaded here (https://osf.io/wcbev/).

## METHOD

### Types of studies and sample characteristics

The seven studies included in the present manuscript were carried out by our research group and employed cross-sectional quasi-experimental designs. To the best of our knowledge, there are no other studies so far comparing groups as a function of physical fitness and/or type of sport practice, that used the same task (i.e., PVT) as a vigilance measure and reported $VO_2$ as an index of cardiovascular fitness, both essential requirements to be included in the analysis. A total sample of 361 participants (114 females) of an age range between 10 and 50 years old, different levels of cardiovascular fitness (i.e., high-fit vs. low-fit) and sport type participation (i.e., self-paced vs. externally-paced exercise) were included in the present multi-study analysis (Table 1). All participants reported normal or corrected-to-normal vision, had no history of neurological problems or cardiovascular diseases, and were not taking any medications that may affect cognitive functions.

### Sport type characteristics and $VO_2$ estimation method

Participants included in the externally-paced group practiced sport modalities such as football, basketball, volleyball, tennis or martial arts, while participants included in the self-paced group practiced sport modalities such as track & field, running, swimming, triathlon or cycling. Importantly, all participants included in sedentary groups (i.e., low-fit groups) reported no historical participation in any sport and were not physically active (less than 2 h per week).

Across studies, $VO_2$ consumption ($ml \cdot kg^{-1} \cdot min^{-1}$) during exercise was employed as the main index of cardiovascular fitness and was estimated and reported using three different methods: A = estimation of the $VO_{2max}$ ($ml \cdot kg^{-1} \cdot min^{-1}$) from the maximum power output measured in watts in a maximal incremental cycle-ergometer test (*American College of Sports Medicine, 2013*) (Studies 1 and 7; see *Luque-Casado et al. (2013)* and *Ballester et al. (2019)* for details); B = estimation of the $VO_{2max}$ ($ml \cdot kg^{-1} \cdot min^{-1}$) from the Léger Multi-stage fitness test (*Léger et al., 1988*) (Studies 2 and 6; see *Ballester et al. (2015, 2017)* for details); C = direct measure of oxygen uptake ($ml \cdot kg^{-1} \cdot min^{-1}$) at the ventilatory anaerobic threshold (VAT; $VO_{2 \, at \, VAT}$) in a submaximal incremental cycle-ergometer test (Studies 3–5; see *Ciria et al. (2017)* and *Luque-Casado et al. (2016b)* for details).

In order to obtain a single measure of cardiovascular fitness across studies, we standardized the $VO_{2max}$ (or $VO_{2 \, at \, VAT}$) data for each participant using the corresponding sedentary group as reference (i.e., the participant's $VO_2$ minus the mean $VO_2$ of the sedentary group (in the same study) divided by the unbiased-populational-$VO_2$ standard deviation estimated from the same sedentary group). Henceforth, we will refer to this standardized measure as simply $VO_2$, which is interpreted as a measure of individual (higher or lower) cardiovascular fitness, relative to same age range and similar sociodemographic extraction peers who do not exercise regularly. Extra measures to

prevent different $VO_2$ estimation methods to bias results will be described in the statistical analysis and results sections.

## Psychomotor vigilance task

A modified version of the PVT developed by *Wilkinson & Houghton (1982)* was used in all studies included in the multi-study analysis. This task was designed to measure sustained attention by recording participants' RT to visual stimuli that occur at random inter-stimulus intervals (*Loh et al., 2004*; *Basner & Dinges, 2011*). The PVT is a simple and reliable task to measure vigilance given the monotonous, repetitive, and unpredictable nature of the target onset (*Drummond et al., 2005*). In the standard procedure, a black circle with a red edge ($6.68° \times 7.82°$) is displayed at the center of the screen in a black background. Later, in a random time interval (from 2,000 to 10,000 ms), the circumference begins to be filled in a red color and in a counter-clockwise direction with an angular velocity of 0.094 degrees per second. The participants are instructed to respond as fast as they can to stop it. They must respond with their dominant hand by pressing the space bar on the PC. Feedback of the response time is displayed on the screen on each trial for 300 ms. The next trial begins after 1,500 ms. Response anticipations are considered as errors. Participants are allowed 3,750 ms to respond. If a response is not made during this time, the message "You did not answer" appears on the screen.

Different task durations (5, 9, 10, 12 and 60 min) and characteristics of the stimuli (gabor patch or red circumference) were used according to the necessary adaptation to the aims of each study (Table 1). In any case, the original paradigm of the PVT task was always maintained making possible the comparison between studies. In extended task durations (i.e., 60 min), we observed a differentiated pattern of RT performance between groups of participants as a function of the time-on-task (*Luque-Casado et al., 2016b*). Therefore, in order to maintain homogeneity between studies, we only analyzed the RT from the first 5′ of the task based on three main reasons: (1) this duration corresponds to the shortest version of the PVT we have used in all studies; (2) given that the Group and Time-on-task factors seem to interact in extended task durations, the selection of the first 5 min (when the effects of greater magnitude were observed and these do not depend on the time-on-task) reduces the complexity of the statistical model; (3) 5 min are sufficient for a reliable measure of vigilance (*Loh et al., 2004*). As a common premise for the studies included in the analysis, the experimental session was administered alternatively between morning or afternoon hours among participants of each group except for studies including children (i.e., study 2 and 6), which were carried out during the afternoon due to hourly restrictions of the participants.

## Design and statistical analyses

Reaction times from the seven studies conducted in our laboratory were collapsed into a single dataset to estimate the effects of sport type (i.e., self-paced, externally-paced and sedentary conditions) on them. In face of the diversity of samples' characteristics and study features, we fitted RTs using multilevel linear mixed-effects modelling, as implemented in the *lme4* R package (*Bates et al., 2015*).

MULTILEVEL DATA STRUCTURE

**Figure 1 Schematic representation of the multilevel data structure.**

**Table 2 Models, fitting indices, and likelihood ratio comparisons.**

| Model | Fixed part | df | AIC | L. ratio | p | |
|---|---|---|---|---|---|---|
| *Baseline* | Age + $VO_2$ + trial | 9 | 34,881.04 | | | |
| *$H_1$* | baseline + sport type | 11 | 34,866.47 | 18.569 | <0.001 | >baseline |
| *Interaction 1* | saturated − (sport type × age) | 15 | 34,865.81 | 8.089 | 0.088 | <saturated* |
| *Interaction 2* | saturated − ($VO_2$ × age) | 17 | 34,863.70 | 1.977 | 0.372 | = saturated |
| *Interaction 3* | saturated − (sport type × trial) | 17 | 34,863.59 | 1.868 | 0.393 | = saturated |
| *Saturated* | $H_1$ + (sport type × age) + ($VO_2$ × age) + (sport type × trial) | 19 | 34,865.72 | | | |
| *Best-fitting* | $H_0$ + sport type + (sport type × age) | 15 | 34,861.58 | 31.465 | <0.001 | >baseline |

**Notes:**
All models have been adjusted with the Maximum Likelihood (ML) method. Age effects include a linear and a quadratic component, jointly included/excluded for model comparisons. The random part is common to all models (see text).
">" Indicates better fit.
"<" worse fit.
"=" not substantially worse fit.
* For factor inclusion/exclusion comparison, a relatively lenient $p < 0.010$ significance level has been used.

In order to take into account the dependencies potentially generated by any procedural differences between studies, we treated RTs as obeying to a multilevel data structure (Fig. 1), with participant (level 3), nested into study (level 2), and study nested into estimation method (level 1). Thus, the random part of the model consisted of intercepts for participant (nested in), study (nested in), $VO_2$ estimation method. This random part was common to all models. The fixed part in a first, *baseline model* ($H_0$) consisted of $VO_2$, age and trial number (to facilitate model convergence, gender was not included as a fixed factor, as it is mostly controlled for by participant and $VO_2$, and not included in any further interactions). Additionally, the effect of age consisted of a linear and a quadratic component, to allow for non-linearity in the age-RT association. This baseline model was pitched against a second, *$H_1$ model* with the same random and fixed parts as $H_0$, plus sport type (self-paced, externally-paced, sedentary) as an added fixed-effects factor. A third, *saturated* model further included the sport type × age, the $VO_2$ × age, and the sport type × trial interactions (please, note that interactions involving age actually refer to two different interaction effects, one involving the linear component and the other involving the quadratic component of the age effect, that were always jointly included/excluded for model comparisons). Interactions were later removed one-by-one (*Interaction 1*, *Interaction 2* and *Interaction 3* models; see Table 2). If any of the three interactions

**Table 3 Effect estimates (B), standard errors (SE), and significance levels for all fixed effects in the best-fitting model.**

| | Full dataset | | | | Restricted dataset | | | |
|---|---|---|---|---|---|---|---|---|
| | **B** | **SE** | **t** | **p** | **B** | **SE** | **t** | **p** |
| Intercept | −0.047 | 0.231 | −0.202 | 0.840 | 0.356 | 0.544 | 0.654 | 0.513 |
| Trial | 0.044 | 0.006 | 8.001 | <0.001*** | 0.069 | 0.009 | 8.145 | <0.001*** |
| VO$_2$ | −0.018 | 0.016 | −1.157 | 0.247 | −0.012 | 0.038 | −0.316 | 0.752 |
| Age (linear) | −24.731 | 9.549 | −2.590 | 0.010** | −16.066 | 16.541 | −0.971 | 0.332 |
| Age (quadratic) | −5.087 | 8.515 | −0.597 | 0.550 | 2.671 | 8.961 | 0.298 | 0.766 |
| Sport type (C1) | −0.071 | 0.026 | −2.667 | 0.008** | −0.097 | 0.029 | −3.318 | 0.001*** |
| Sport type (C2) | 0.024 | 0.035 | 0.696 | 0.487 | 0.032 | 0.058 | 0.560 | 0.577 |
| Age (linear) × sport type (C1) | 1.422 | 7.381 | 0.193 | 0.847 | 5.647 | 2.310 | 2.445 | 0.015* |
| Age (quadratic) × sport type (C1) | −7.877 | 5.671 | −1.389 | 0.165 | −1.850 | 2.622 | −0.706 | 0.481 |
| Age (linear) × sport type (C2) | −3.673 | 3.037 | −1.209 | 0.227 | −2.171 | 4.782 | −0.454 | 0.651 |
| Age (quadratic) × sport type (C2) | 0.413 | 2.972 | 0.139 | 0.890 | 2.038 | 5.765 | 0.354 | 0.724 |

**Notes:**
The best fitting model was adjusted with the REML method. The selected model was fitted and run with the full dataset, and subsequently also run with a restricted dataset including the studies in which there were participants in the three sport types. C1 represents the contrast between the externally paced and the other two sport types. C2 represents the contrast between externally paced and self-paced sport types.
\* $p < 0.05$.
\*\* $p < 0.01$.
\*\*\* $p < 0.005$.

contributed to the model fit, it was kept in the final, *best-fitting* model. This last, best-fitting model thus consisted of the same random and fixed parts as the H$_1$ model plus the interactions identified as substantially contributing to model fit. This model was used to estimate effects.

Prior to fitting, RTs larger than 1,000 ms (i.e., omission errors; 0.5% trials) were removed from further analyses, and the remaining ones (16,239) were log-transformed. To enable model convergence and facilitate interpretation of regression coefficients, trial number and log-transformed RTs, were scaled and zero-centered. Models were compared using the Akaike Information Criterion (AIC), and a Likelihood ratio test. For model comparisons performed to identify the best-fitting model, a relatively lenient 0.010 *p*-value criterion was adopted. Inferences were subsequently made based on the results of the best-fitting model.

## RESULTS

Table 2 shows the fitting indices for all the models included in the relevant comparisons described above (baseline, H1, saturated, interaction 1–3, and best-fitting models).

Table 3 (columns 1–5) displays unstandardized regression coefficients (B), their standard errors and significance levels (according to *t*-tests), for each fixed component in the best-fitting model. The best-fitting model included the effects of sport type, and the sport type × age interaction.

In the best-fitting model, the contrast corresponding to the comparison between externally paced sports and the other two conditions (sedentary and self-paced sports, pooled together) yielded a significant *t*-test (contrast 1); whereas self-paced sports did not differ from the sedentary condition (contrast 2). Neither C1 nor C2 interacted with either the linear or the quadratic effect of age.

The absence of a significant effect of $VO_2$ in the best-fitting model ($p = 0.247$) was also confirmed by an approximated Bayes Factor (*Wagenmakers, 2007*). In order to compute it, we built a third model ($H1_b$), equivalent to the best-fitting model, but with $VO_2$ removed from the fixed part. The AIC value for this model was 34,861.02. The Bayes Factor approximation for this comparison between the best-fitting model and $H1_b$ was $BF_{10} = 0.016$, which supports $H0_b$ relative to the best-fitting model, and suggests the absence of any substantial influence of $VO_2$ on RTs if sport type is controlled for.

The upper panel of Fig. 2 displays predicted (standardized) RTs across age for the three sport types. Although the age $\times$ sport type interactions were not significant, the effect of sport type seems to vanish for intermediate ages. Additionally, given there were virtually no practitioners of externally paced sport among 40–50 years-old participants, predictions for that age group mostly resulted from fitting the effect of age on RT for younger participants. In order to ensure the effect of sport type was not inflated in the previous analysis, we run the best-fitting model with a dataset restricted to the studies in which there were participants in the three sport types (study 6 and 7). As shown in columns 6–9 in Table 3, the effect of sport type remained largely significant, but there also was an interaction between the sport type C1 contrast and the linear component of age. The lower panel of Fig. 2 shows the shape of this interaction. Importantly, qualitative predictions for these studies were virtually identical to the ones made from the full dataset for the 10–35 age range.

## DISCUSSION

Sedentarism has been related to numerous health issues and, according to a wealth of literature, to poorer cognitive function (with respect to active individuals) (*Gomez-Pinilla & Hillman, 2013*). The aim of this paper was to contribute to that body of work by pooling the data from seven studies ($N = 361$) to examine whether $VO_2$ (index of cardiovascular fitness), type of sport participation (externally-paced and self-paced), or both (controlling for sex, age and the method of $VO_2$ estimation), are crucial factors to explain the association between the regular practice of exercise and vigilance capacity (measured by means of the PVT).

The results were straightforward. Sport type was significantly related to RT, although only externally-paced sport differed from the sedentary condition (and also from the self-paced sport condition). Sport type and age interacted, showing that the (same) pattern of differences between sport types was more evident in children and adolescents than in older participants. Both the multilevel linear mixed-effects modelling and Bayesian analysis confirmed the absence of effect of $VO_2$ and self-paced sport practice (with respect to the sedentary condition) on RT.

Maximum oxygen consumption ($VO_{2max}$) has been the primary index of cardiovascular fitness to associate with cognitive and brain functioning (and anatomy) over the last years. The positive findings to date establish that the greater the $VO_{2max}$, the higher the cardiovascular fitness and the better the cognitive and brain functioning (*Gomez-Pinilla & Hillman, 2013*; *Åberg et al., 2009*; *Tarumi & Zhang, 2015*; *Voss et al., 2016*). Together with the outcome of randomized controlled trials (RCT; where increased $VO_{2max}$ after
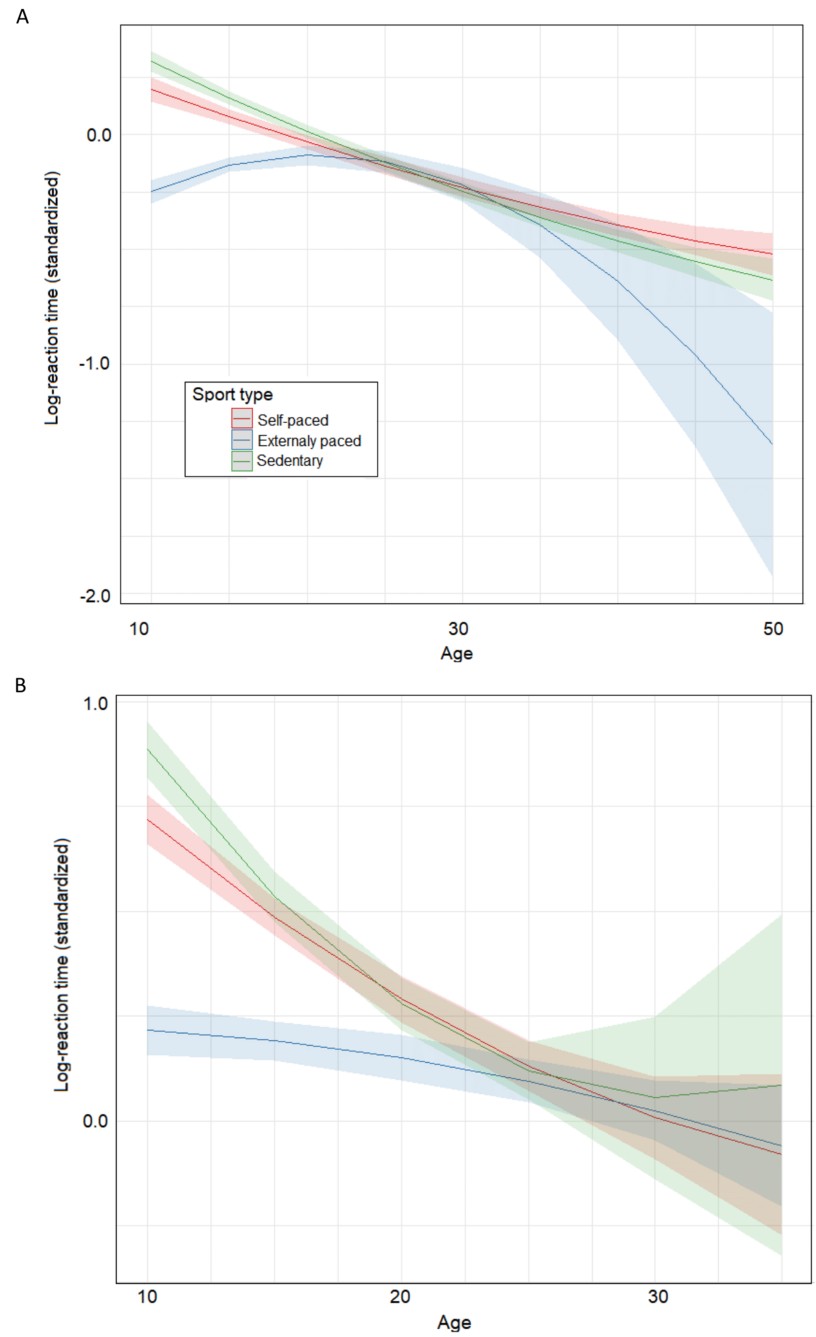

**Figure 2 Predicted effect of sport type across age.** (A) Predicted (standardized) reaction times (RTs) across age for the three sport types including the full dataset; (B) predicted (standardized) RTs across age for the three sport types including only dataset restricted to the studies in which there were participants in the three sport types (study 6 and 7). Figure shows that externally-paced sport practice (e.g., football) entail significantly shorter RT compared to self-paced sport (e.g., triathlon) and sedentary condition irrespective of the cardiovascular fitness level, depicting larger effects in children and adolescents.

exercise intervention was accompanied by enhanced cognition) (*Erickson et al., 2011*) this evidence has fueled the cardiovascular (selective) hypothesis, that is, the regular practice of cardiovascular exercise (not other forms of physical activity such as stretching;

cf. *Kramer et al., 1999*) positively affect cognition (with a selective influence on executive function) by means of its physiological effects at neural level. The null effect of $VO_2$ and the non-significant RT difference between self-paced exercise and the sedentary condition (well differentiated in terms of cardiovascular fitness) in our study appear to challenge that hypothesis (cf. *Etnier et al., 2006*).

A simple explanation of our null result is that $VO_{2max}$ is a measure of cardiovascular fitness that is not sensitive enough to variations in cognitive performance (RT in the PVT here), even though, as we said above, it is the main index used to date. Proponents of the cardiovascular selective hypothesis could also argue that our null finding was due to the low executive demands of the PVT. The evidence from neuroimaging data speaks against that account. There is indeed work reporting that PVT performance involves selective activation of brain areas related to sustained attention and cognitive control (*Drummond et al., 2005*; *Zhu et al., 2018*), presumably driven by the (large) temporal uncertainty of the target appearance, the repetitive and monotonous nature of the task and the need of generating temporal expectations. Of course, in defense of the cardiovascular selective hypothesis, it could still be claimed that the executive demands of the PVT are much lower than those of the conflict or working memory tasks used in previous research on this topic. This is a matter open to opinion and debate as no study to date has addressed this issue directly.

The fact that self-paced exercise was not related to improved RT with respect to the sedentary condition is not surprising if one considers that only half of the studies in which we compared a group of self-paced sport athletes with a group of sedentary individuals revealed statistically significant RT differences. Assuming that the large $N$ in the present analysis ensures sufficient statistical power, one could argue that those reported significant group RT differences were false positives (at least considering the first 5′ of the task). However, those RT group differences from the single studies might still be meaningful but explained by the influence of an unknown (uncontrolled) variable. Moreover, in the studies showing positive results, group differences were not only seen in terms of RT but also in accuracy performance in an oddball task (with much lower RT demands than the PVT; see *Ciria et al., 2017*), and, more importantly, in task-related cardiac and electroencephalographic measures (*Luque-Casado et al., 2013*, *2016a*, *2016b*). If those group differences were to be true, the result of the present analysis suggests that they were not due to mere differences in cardiovascular fitness.

Even if it involves executive functioning, the PVT is clearly a task demanding visuomotor coordination to react as rapidly as possible to the target appearance. This could explain that only externally-paced sport practice associates to (enhanced) RT performance in our study, supporting the cognitive skills hypothesis whereby sport practice is just another medium for cognitive training (*Voss et al., 2010*), over and above its effect due to cardiovascular physiological adaptations. Obviously, this result cannot be taken as evidence of the *effect* of externally-paced sport practice on the vigilance capacity of our participants for other variables not related to the sport practice itself could well account for the reported positive relationship. For instance, pre-existing individual differences that biased the choice of the particular sport practice (*Belsky et al., 2015* for a related

argument) might explain the sport-type effect in our multi-study analysis, and also the interaction between sport type and age. In fact, most of the younger participants in the externally-paced sport type groups were football players from (two) Spanish 1st Division League junior teams (with strict selection criteria and talent identification programs) while older participants were amateur/recreational athletes. However, the above and any other alternative explanation are speculative, for only well-designed RCTs would establish cause-effect relationships between exercise practice and cognitive performance. Also, future research should clarify whether indexes of response accuracy (beyond psychomotor response speed typical of the PVT) from other cognitive tasks measuring sustained attention (e.g., oddball task) discriminate between the type of sport practice and cardiovascular fitness in relation to the ability to maintain attention over time.

In the absence of that RCT, one could still argue that the extant evidence on the effect of chronic exercise on cognition supports the hypothesis that practicing exercise and sport regularly would have a positive effect on vigilance capacity. However, that evidence is not as conclusive as it may appear, with some systematic reviews and meta-analysis reporting positive results (*Colcombe & Kramer, 2003*; *Guiney & Machado, 2013*) and other showing null effects (*Angevaren et al., 2008*; *Singh et al., 2018*; *Verburgh et al., 2014*; *Young et al., 2015*).

## CONCLUSIONS

The results of the multistudy analysis reported here point to the type of sport practice as a major factor to explain differences in vigilance performance as a function of regular exercise over and above the level of aerobic fitness, which in turn challenges the cardiovascular hypothesis. In any case, this topic warrants further well-designed research (RCTs) to unveil whether chronic exercise (and sport practice) has a true effect on cognition in general, and vigilance in particular. Interested researchers are facing a challenging task, as they would have to take into account the cognitive demands of the sport (exercise) activity as we highlighted here, the many other factors related to exercising (such as the intensity and duration of each exercise session or the periodization of the exercise program) and all potential mediators (BDNF, sleep, motivation, etc.) (*Stillman et al., 2016*). Only after solid evidence is acquired, researchers would be ready to pave the road for prescription of exercise as a potential tool to enhance cognition and prevent cognitive decline.

## ACKNOWLEDGEMENTS

We thank to all the participants who took part in the experiment.

### Funding

This research was supported by a postdoctoral grant from the Spanish "Ministerio de Ciencia, Innovación y Universidades" (FJCI-2016-28405) to Antonio Luque-Casado, predoctoral grants from the Spanish Ministerio de Economía, Industria y Competitividad

to Luis F. Ciria (BES-2014-069050), and to Rafael Ballester (FPU13-05605), and research grants from the "Ministerio de Economía, Industria y Competitividad" (PSI2013-46385-P and PSI2016-75956-P) and the "Junta de Andalucía" (SEJ-6414) to Daniel Sanabria. The funders had no role in study design, data collection and analysis, decision to publish, or preparation of the manuscript.

### Grant Disclosures

The following grant information was disclosed by the authors:
Spanish "Ministerio de Ciencia, Innovación y Universidades": FJCI-2016-28405.
Spanish Ministerio de Economía, Industria y Competitividad to Luis F. Ciria: BES-2014-069050.
Rafael Ballester: FPU13-05605.
Ministerio de Economía, Industria y Competitividad: PSI2013-46385-P and PSI2016-75956-P.
"Junta de Andalucía": SEJ-6414.

### Competing Interests

The authors declare that they have no competing interests.

### Author Contributions

- Daniel Sanabria conceived and designed the experiments, analyzed the data, authored or reviewed drafts of the paper, approved the final draft.
- Antonio Luque-Casado performed the experiments, prepared figures and/or tables, authored or reviewed drafts of the paper, approved the final draft.
- José C. Perales conceived and designed the experiments, analyzed the data, contributed reagents/materials/analysis tools, prepared figures and/or tables, authored or reviewed drafts of the paper, approved the final draft.
- Rafael Ballester performed the experiments, approved the final draft.
- Luis F. Ciria performed the experiments, approved the final draft.
- Florentino Huertas conceived and designed the experiments, approved the final draft.
- Pandelis Perakakis conceived and designed the experiments, approved the final draft.

### Data Availability

All data and R script can be found at the Open Science Framework: Sanabria, Daniel, Antonio Luque-Casado, José C. Perales, Luis F Ciria, and Pandelis Perakakis. 2019. "The Relationship between Vigilance Capacity and Physical Exercise: A Mixed-Effects Multistudy Analysis." OSF. April 9. DOI 10.17605/OSF.IO/WCBEV.

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
