# Peer review of "The relationship between vigilance capacity and physical exercise: a mixed-effects multistudy analysis"

_PeerJ, doi:10.7717/peerj.7118_

## Round 0.1 · original submission · Major Revisions

I would recommend to pay particular attention to the issues brought forward by Reviewer 1. It is reasonable to expect that the reader should be confident that the papers selected were not simply those of convenience (essentially collapsing the authors own data) but have been chosen in a systematic way. Please provide more details as to the inclusion/exclusion rationale, search terms, databases, and selecting and screening studies. it would be helpful to show, using a table or figure, how the final group of studies were decided upon.

Reviewer 1 ·

Basic reporting

The purpose of this article was to provide a systematic, quantitative review of the literature depicting the relationship between physical activity and cognition as measured by reaction time as part of the Psychomotor Vigilance Task. To accomplish this objective, the author(s) compiled 7 unique studies that were purported to assess the aforementioned relationship comparing type pf physical activity (i.e., externally paced, self-paced, vs. sedentary behavior) while controlling for age, sex, and type of VO2 test. A multilevel linear mixed-effects model was employed to assess this relationship.

Overall the manuscript is well constructed and conforms to the basic standards. However, initially the narrative leads the reader to believe that the author(s) have done an extensive search of the extant literature prior to conducting their analysis. Yet later in the manuscript it appears that the author(s) are simply providing a systematic review of their own respective body of work. The author(s) need to address this issue, be clear and consistent throughout. Furthermore, several grammatical and structural issues are consistent throughout. Please see specific comments below:


Abstract
1. P1 line 21-35: The use of parentheses is often misused and disrupts the flow of the manuscript. For example, the author(s) state: Further analyses confirmed the absence of effect of VO2 and self-paced sport practice (with respect to the sedentary condition) on RT. Please consider removing the parentheses and rewriting the statement for clarity: “With respect to the sedentary condition, further analyses confirmed the absence of effect of VO2 and self-paced sport practice on RT.
2. P1 line 33. Please consider replacing the term “expertise” with a more participatory term such as “participation” or “involvement”. The use of the term “expertise” is overstated here and throughout the manuscript unless the author(s) can provide greater rationale and detail regarding participant characteristics and years of experience.


Introduction
1. Please use page numbers throughout.
2. P1 line 46-47: Please add parentheses around text: (i.e., the ability to stay focused, and to detect and respond efficiently to target stimuli in order to attain the goals of the task).
3. P1 line 50: Replace attain with accomplish.
4. P1 line 55: Please replace this sentence “The enhanced physical or cardiovascular fitness is one of the main consequences of the regular practice of physical exercise that has been related to cognitive performance” with “Enhanced physical or cardiovascular fitness is one of the main consequences of regular practice of physical exercise that has been related to cognitive performance”.
5. P1 line 51-53. Consider revising the current sentence to read: …that used a measure of reaction time (RT) from the Psychomotor Vigilance Task (PVT) consistently used by our research group (ADD REFERENCE).
6. P2 line 61: Delete “the”
7. P2 line 62: Delete “the”
8. P2 line 63: Add reference to the end of the sentence.
9. P2 line 64: Replace “Exercise practice” with “Physical activity”
10. P2 line 64: Awkward. Please consider revising sentence to read: “Given the variety of physical activities available, exercise is far more than just a way for enhancing cardiovascular fitness (ADD REFERENCE). In fact, given the varying perceptual and cognitive demands of physical activity it is expected that cognitive enhancement would follow sustained practice (ADD REFERENCE).
11. P2 line 68: Whenever using i.e., or e.g., please add parentheses. Please amend the remaining document.
12. P2 line 72: Replace “cognitive skills demands” with “cognitive demands”
13. P2 line 73: Please add reference after “research”.
14. P3 line 83: Replace “This is the…” with “For this…”
15. P3 line 83: Delete “why” and add “have” after we. The sentence should read: For this reason, we have decided to perform a mixed-effects analysis on data from 7 unique studies involving 361 participants.
16. P3 line 89: Replace the use of “’” with “minutes”
17. P3 line 95: Consider revising this sentence to read: “In brief, this report aims to advance our current understanding of the research linking physical activity and cognition, with a focus on vigilance.

Methods


1. P5 line 135: Add parentheses before and after i.e.,
2. P5 line 133-142: Please consider rephrasing this paragraph. There are considerable run on sentences. This section is difficult to interpret as currently written.
3. P5 line 147: Delete “(vigilant)”

Experimental design

1. P3 line 101. Ethics Statement. Given that this research is using previously published research as data, the inclusion of an ethics statement is not necessary. For example, the author(s) would not have consented participants for inclusion in this quantitative review since they are using preexisting and previous published data. As a result, please consider removing this section.

2. P4. Line 109. Types of Studies and Sample Characteristics. The author(s) should consider providing a greater rationale and description of the inclusion criteria for articles retained in the review. Equally, the author(s) should describe the search strategies used to obtain said articles. For example what search engines were used? Search terms? Did the author(s) use an ancestry or descendancy approach to search the existing literature? How many articles were unveiled vs. retained? Please be as detailed as possible.

3. P4. Line 109. Types of Studies and Sample Characteristics. In the section, the author(s) again use the term expertise. As it currently stands this is a fatal flaw in the rationale for the subsequent analyses given that no information is provided indicating any level of “expertise” in a given activity. At best, the author(s) have identified participants varying in the type of activity frequented but not in varying levels of expertise. Please alter the language around ‘expertise” or provide greater detail regard participant characteristics warranting the use of the term expert. Please correct throughout the manuscript.

4. P6 line 172. Design and Analysis. It is unclear where the specific data points that were collapsed into a single dataset have come from. This appears to arise from a lack of clarity surrounding the inclusion criteria and how the author(s) access the raw data from the studies included. A more thorough description is needed here.

5. I commend the author(s) for their attempt to employ higher level statistical analyses. However, the more complex the analysis the more removed we become from a practical interpretation of the results. Given the access to raw data, and the ability of the author(s) to normalize the data, I would encourage the author(s) to consider a much simpler analysis. For example, an ANOVA comparing externally paced, self-paced, and. sedentary behavior on RT. Equally, given the interest in the role of VO2 on cognitive performance an ANCOVA can be equally employed.

Validity of the findings

Although I believe the initial research question is of interest to the readership of Peer J, several practical, methodological, and inferential decisions lead me to question the overall utility of the article and its ability to advance our understanding of the relationship between fitness level, cognition and activity type. For example, the author(s) suggest that children who practice externally-paced tasks outperform those who engage in self-paced tasks suggests that simply looking at RT as a dependent variable limits the insights gleaned from such an investigation, as such the authors should consider including on their analysis the number of errors during the test. In other words, does activity type and fitness level influence not only speed but accuracy as well? As a result, the methodological approach to this investigation is called into question and warrants further detail and elaboration.

Results appear mixed given the different types of tests chosen. Can the author(s) justify drawing a positive conclusion based on the AIC results while ignoring those of the BIC? In other words, it appears that the author(s) report results using AIC in some situations and BIC in others. I would encourage the author(s) to use and report both approaches with each analysis and then later offer an explanation for any similar and/or disparate results; including the implications of disparate results.

Additional comments

There are sections of the manuscript that are clear, concise, and well written. However, much of the manuscript is need of significant re-write due to formatting, grammatical, and punctuation errors. Addressing these issues would significantly enhance the article.

Reviewer 2 ·

Basic reporting

no comment

Experimental design

1. In my opinion, the study 3 should not be included in the analyses because there was no participants who belong to Extremely-paced sports in the middle age group.
2. The exercise effect might be different between age groups. Thus, the authors should consider the interaction effect of age groups (adolescent, young adults) × aerobic fitness, and age groups × sports types.
3. Individual PVT performance might be change in the day. Thus, it should be described what time the participants conducted the task.
4. There is no explanation why RTs larger than 1000 ms were removed from the analyses.
5. With the exception of the 7 studies which was included in the analysis, the authors published the other paper which could be included the analyses (Luque-Casado, A. et al., Scientific report, 2016). The authors should describe the criteria to include the analyses.

Validity of the findings

There is no description about random slope effect of task trial number.

Additional comments

In this study, using previous studies, the authors performed mixed effects multistudy analysis to examine whether cardiovascular fitness, sports-related expertise, or both are the key factors associating to the vigilance performance. In the results, they found only externally-paced sports was associated with PVT performance. The results is interesting and might sheds light on how physical activity influence vigilance performance. However, the authors have revealed that participants who engage self-pace sports have higher vigilance performance than sedentary participants in several studies. It is difficult to think these positive effects of self-paced sport was false-positive results due to small sample size. The authors should carefully discuss the difference between studies which showed positive effects of self-paced sport and studies which showed null effects from the aspect of the experimental design, task design, and participants’ characteristics.

·

Basic reporting

This brief report presents a mixed-effects analysis including data from seven studies (total N=361) that used the same measure of vigilance capacity. This manuscript is particularly clear and well written. The state of art is update and from my point of view the data analysis seems relevant.

Experimental design

This study, within ams and scope of the journal, aimed to examine whether cardiovascular fitness (VO2) and/or type of sport expertise (externally-paced and self-paced) could explain the association between the regular practice of exercise and vigilance capacity (reaction time RT from the first 5’ of a Psychomotor Vigilance Task).

Validity of the findings

The null effect of VO2 and the non-significant RT difference between self-paced exercise and the sedentary condition (well differentiated in terms of cardiovascular fitness) did not support the cardiovascular fitness hypothesis assuming that increased VO2, induced by regular exercise, is responsible for cognitive improvement. Interestingly, the present findings suggest that only externally-paced sport practice (which required rapid adaptation and response to the constantly varying exercise environment) affects vigilance in particular. The results of the present study open new perspectives related to cognitive improvement related to sport expertise, and to the potential benefit effect of regular perceptual and cognitive solicitations induced by physical activities.

Additional comments

I would like to congratulate the authors for this particularly noteworthy work and for the innovative perspectives that this study offers in the field of exercise and cognition. I am favorable to the publication of the present article in his current form without modifications.

---

## Round 0.2 · Minor Revisions

The paper is much improved. There is some clarity needed in reporting of the findings in order to make the messages translatable.

Line 25 Use of terms “externally paced- self paced” in abstract is without context. I would provide examples here.

Lines 29-33: The abstract should be self-contained and informative. The phrases “significantly related to” and “absence of effect of” actually do not provide the reader with a tangible message. Please state exactly the findings. For example, “externally-paced sports (examples of these) resulted in significantly faster RT but mainly in children and adolescents”.
Results: Figures are informative but as discussed above, more context is required. For example, the authors could calculate %change in RT for a specific age group based on the models. Informative statements could be framed, such as “Among 10 year old children, for every 5min of exercise while playing soccer there was a 2% faster RT when compared to treadmill running, irrespective of cardiovascular fitness”.

·

Basic reporting

I have no major complaints.
Page 12, line 263/264: an unnecessary bracket need to be deleted.
Page 17, line 282: "electroencephalografic" should be corrected.

Experimental design

no comment

Validity of the findings

I think that the present results are interesting and worth sharing.

---

## Round 0.3 · accepted · Accept

The authors have adequately addressed all the concerns raised.